# New Generation of Systemic Inflammatory Markers for Respiratory Syncytial Virus Infection in Children

**DOI:** 10.3390/v15061245

**Published:** 2023-05-25

**Authors:** Omer Okuyan, Yusuf Elgormus, Seyma Dumur, Ugurcan Sayili, Hafize Uzun

**Affiliations:** 1Department of Child Health and Diseases, Medicine Hospital, Istanbul Atlas University, 34408 Istanbul, Turkey; 2Department of Medical Biochemistry, Faculty of Medicine, Istanbul Atlas University, 34408 Istanbul, Turkey; 3Department of Public Health, Cerrahpasa Faculty of Medicine, Istanbul University-Cerrahpasa, 34303 Istanbul, Turkey

**Keywords:** respiratory syncytial virus, lower respiratory tract infection, systemic immune–inflammatory index, neutrophil–to–lymphocyte ratio, platelet–to–lymphocyte ratio, C-reactive protein

## Abstract

Aim: This study evaluated the relationship between the systemic immune–inflammatory index (SII), neutrophil–to–lymphocyte ratio (NLR), and platelet–to–lymphocyte ratio (PLR) with clinical findings of respiratory syncytial virus (RSV) infection among children with a diagnosis of lower respiratory tract infection (LRTI). Methods: The study was conducted between 1 January 2020 and 1 January 2022 in a pediatric clinic. This retrospective study included 286 consecutive patients between 0 and 12 years of age, 138 of whom were RSV (+) (48.25%) and 148 of whom were RSV (−) (51.75%). The detection of the RSV antigen was carried out using the chromatographic immunoassay method on nasopharyngeal swabbing samples. Results: CRP content was significantly higher in patients with RSV (+) than in children with RSV (−), while NLR, PLR, and SII, as inflammatory parameters, were significantly lower. Fever, coughs, and wheezing were the most common symptoms in the RSV (+) groups (100%). RSV infections were the highest in November, October, and December, in that order. The AUC was statistically significant for parameters in all groups. AUC values were 0.841 (95%: 0.765–0.917) for leukocytes, 0.703 (95%: 0.618–0.788) for lymphocytes, 0.869 (95%: 0.800–0.937) for CRP, 0.706 (95%: 0.636–0.776) for NLR, 0.779 (95%: 0.722–0.836) for PLR, and 0.705 (95%: 0.633–0.776) for SII. CRP was found to have both high sensitivity (80.4%) and high specificity (82.4%) among all parameters. While the ROC analysis results showed similar results for children under two years old, only CRP and NLR were statistically significant in this group. Conclusion: CRP performed better than other blood parameters as a marker. The NLR, PLR, and SII index were significantly lower in LRTI patients with RSV (+) than in those with RSV (−), which implies a higher grade of inflammation. If the cause of the disease can be determined by this method, disease management will be easier, and unnecessary antibiotics could be avoided.

## 1. Introduction 

Respiratory syncytial virus (RSV) is the most common and most important cause of viral lower respiratory tract infections (LRTI) in infants and children worldwide. RSV should be considered during the differential diagnosis of patients of any age with signs of LRTI [1]. RSV infection is frequently seen in children under two years of age. Nearly half of all infections progress from the upper respiratory tract to the LRT, and approximately 2% of patients need to be hospitalized for this reason [2,3]. Up to 50–70% of infants aged one year and 95% of all infants up to two years of age are infected with RSV. Despite the development of serum antibodies against RSV in years following infection, reinfections with RSV may develop [3,4]. There are also general risk factors independent of age and person, including low socioeconomic status, crowded living conditions, inability to breastfeed, malnutrition, smoking in the environment, and family history of asthma or atopy. Risk factors for severe RSV infection include chronic disease, asthma, bronchopulmonary dysplasia (BPD), preterm birth, and congenital heart defects [5]. Patients with these features should also be treated with caution in terms of serious disease.

There is no complete treatment, but supportive treatment can be applied. Preventing hypoxia, regulating hydration, and reducing bronchoconstriction and inflammation are the main lines of treatment. It is also not exactly known which part of the inflammatory cascade that occurs in RSV is affected by treatment. In the current neonatal mouse model of infant RSV disease, the aberrant immune response was accompanied by severe lung injury characterized by marked interstitial and alveolar inflammation, airway mucus production, and eosinophilia [6].

Real-time reverse transcriptase–polymerase chain reaction (RT–PCR) of nasopharyngeal swabs is the current gold standard for testing for RSV, as it requires cost-effective, easily applicable markers. Recently, studies evaluating the roles of complete blood count (CBC) elements and their ratios to each other in determining systemic inflammation and their use as disease activity markers have become quite common due to their low cost and easy accessibility. Among these, parameters such as the neutrophil–lymphocyte ratio (NLR), platelet–lymphocyte ratio (PLR), and mean platelet volume have not been adequately studied in RSV [7]. The systemic immune–inflammation index (SII) is a similar inflammatory marker used as a prognostic predictor in various diseases such as COVID-19 [8]. It is calculated by the following formula: peripheral neutrophil count x platelet count/lymphocyte count.

One in fifty deaths in children 0 to 60 months of age and one in twenty-eight deaths in children 28 days to 6 months of age can be attributed to RSV. RSV passive immunization programs targeting protection during the first six months of life could have a substantial effect on reducing the RSV disease burden [9]. Patients presenting with signs and symptoms of RSV require urgent evaluation, including a detailed history and physical examination focused on pretest probability scores, as well as laboratory tests.

This study aimed to determine a new generation of markers of RSV infection in patients aged 0–12 years who were found to be positive by screening for RSV antigen by taking nasopharyngeal swab samples in the pediatric emergency and pediatric outpatient clinics and diagnosed with LRTI. We aimed to test whether C-reactive protein (CRP), NLR, PLR, and the SII can be used as a preliminary test as an indicator of clinical utility in RSV.

## 2. Material and Method

### 2.1. Study Design and Participants

Ethical approval of this study was obtained by the Non-Interventional Ethics Committee of the Medical Faculty of Istanbul Atlas University (No: E-22686390-050.01.04-14258). The study was performed in accordance with the Helsinki Declaration. The institutional review board that approved the study also waived the need for informed consent due to the retrospective observational nature of the study.

Patients were categorized as either positive or negative, depending on RSV results. A total of 286 consecutive patients between the ages of 0 and 12 years of age were admitted to the Medicine Hospital (pediatric outpatient clinic and pediatric emergency clinic) between 1 January 2020 and 1 January 2022 and were included in the study. Of these 286, 138 were RSV (+) and diagnosed with LRTI, and 148 were RSV (−) and diagnosed with community-acquired pneumonia (CAP) from nasopharyngeal swab samples.

### 2.2. Inclusion Criteria

Children under 12 years old who consulted a physician, i.e., a pediatrician or general practitioner in a pediatric outpatient clinic or pediatric emergency clinic with symptoms of an LRTI, were included in the study. LRTI was defined as the presence of (i) a common cold, (ii) fever, and (iii) any one of cough, congestion or trouble breathing in the last three or six months [10]. Lower respiratory tract inflammation was evaluated with chest auscultation or chest X-ray. RSV positivity (RSV+) was defined as the detection of RSV by the polymerase chain reaction (PCR) method from nasopharyngeal swab samples.

### 2.3. Exclusion Criteria

Infants with a history of recurrent wheezing, those with serious comorbidities (e.g., sepsis, meningitis, etc.), infants with severe neurological and metabolic disorders, children with previously known immune deficiency, those with recurrent RSV infections, and children older than 12 years of age were excluded from the study. Nasopharyngeal swab samples with blood were not included in the study because the presence of blood may affect the test result. Patients with underlying hematologic diseases were excluded. First, RSV infection was determined using a rapid antigen test. After RSV was confirmed using a multiplex PCR kit (INFLUENZA A/B, SARS-CoV-2, RSV), COVID-19 and influenza patients were excluded from the study because the number of patients (n = 3) with COVID-19 was insufficient. Patients diagnosed with acute otitis media with CRP < 5 were excluded.

### 2.4. RSV Diagnosis

The diagnosis of RSV was made using a lateral flow chromatographic immunoassay method with nasopharyngeal swab samples using an RSV test kit (NADAL^®^ RSV Test, Germany). Afterward, RSV was confirmed using the multiplex PCR kit (DIAGNOVITAL INFLUENZA A/B SARS-CoV-2 RSV, Multiplex Real-Time PCR Kit, RTA, Kocaeli/Turkey) on the BIO-RAD CFX device using nasopharyngeal swab samples.

### 2.5. Analysis of Complete Blood Count (CBC)

CBC was recorded with an automatic hematology analyzer (Sysmeks XN-1000, Norderstedt, Germany). NLR and PLR were calculated based on neutrophil/lymphocyte/thrombocyte counts. NLR values were calculated by dividing neutrophil count by lymphocyte count at admission. PLR values were calculated by dividing platelet count by lymphocyte count at admission. SIIs were calculated by the formula neutrophil count x platelet count/lymphocyte count.

### 2.6. Analysis of C-Reactive Protein (CRP)

The serum CRP levels were measured using the nephelometric method (Immage 800 Beckman Coulter, CA 92821, USA). CRP analysis was performed at the time of admission.

### 2.7. Other Variables

The detailed anamnesis of the patients in the study group was taken from their families. Age and gender, complaints at the time of admission, onset time of complaints, number of family members, number of siblings, primary caregiver and care conditions, nutritional history, whether the child has any additional diseases, and findings of physical examinations were recorded as part of the anamnesis.

### 2.8. Statistical Analysis

Statistical Package for the Social Sciences version 21.0 software package for Windows (IBM Corp., Armonk, NY, USA) was used for data evaluation and analysis. Categorical variables are presented as frequencies (n) and percentages (%), and numerical variables are presented as medians (interquartile range). The Kolmogorov–Smirnov test was applied for normality analysis. The chi-square test was used to compare the distribution of categorical variables between groups. The Mann–Whitney U test was used to compare continuous variables between two independent groups. A value of *p* < 0.05 was accepted as statistically significant.

## 3. Results

Table 1 indicates the gender and laboratory findings of RSV-infected and non-infected groups. We found that 52.9% (n: 73) of RSV (+) and 58.8% (n: 87) of RSV (−) were male, and there was no statistically significant difference in terms of gender between the RSV (+) and RSV (−) groups. PLT, neutrophil, and monocyte counts were similar between the RSV (+) and RSV (−) groups. HGB and HCT were statistically significantly lower in the RSV (+) group than the RSV (−) group (*p*: 0.012; *p* < 0.001, respectively). WBC count was higher in the RSV (+) group compared to the RSV (−) group (9.83 (7.53–11.83); 7.6 (6.61–8.39); *p*: < 0.001). The median lymphocyte count was 4.34 (2.65–6.36) in the RSV (+) group and 2.78 (1.4–3.6) in the RSV (−) group. Lymphocyte count was statistically significantly higher in the RSV (+) group than in the RSV (−) group. CRP content was higher in the RSV (+) group compared to the RSV (−) group (4.59 (1.57–13.63); 0.72 (0.55–1.25); *p*: < 0.001). The NLR, PLR, and SII were statistically significantly lower in the RSV (+) group than the RSV (−) group (all of them, *p*: < 0.001). The median NLR was 0.81 (0.41–1.64), the median PLR was 72.17 (49.63–108.26), and the median SII was 243.40 (127.03–554.42) in the RSV (+) group; the median NLR was 1.37 (1.08–2.16), the median PLR was 119.31 (90.32–223.07), and the median SII was 449.5 (349.25–622) in the RSV (−) groups (Figure 1).

Fever, cough, and wheezing were the most common symptoms in RSV (+) groups (100%). Sixteen patients (11.6%) were hospitalized, and the median length of hospital stay was six (three to nine) days, with a minimum of two days and a maximum of 20 days (Table 2).

Figure 2 represents the number of children with RSV infection according to month. RSV infections were highest in November, followed by October and December. RSV infections were low from March to September, increasing after September and peaking in November.

Table 3 and Figure 3 represent the ROC analysis results for the RSV (+) group. The AUC was statistically significant for all eight parameters in all groups (leukocytes, lymphocytes, CRP content, hemoglobin (HGB), hematocrit (HCT), NLR, PLR, and SII). The AUC was 0.869 (95%CI: 0.800–0.937), with 80.4% sensitivity and 82.4% specificity, with a cutoff value of greater than 1.5 for CRP content. The AUCs were 0.706 (95%CI: 0.636–0.776) for NLR, 0.779 (0.722–0.836) for PLR, and 0.705 (0.633–0.776) for SII; specificity values were over 90%. The NLR with a cutoff of lower than 0.85, the PLR with a cutoff of lower than 73, and the SII with a cutoff of lower than 280 all had a specificity of over 90%.

Table 4 represents the laboratory findings of RSV-infected and non-infected patients according to age group. The study group consisted of 237 (82.9%) children under two years of age and 49 (17.1%) children over two years of age. For children under two years of age, WBC, HGB, HCT, lymphocytes, CRP, NLR, PLR, and SII were all found to be significantly different between RSV (+) and RSV (−) groups. However, for children over two years of age, the only significantly different factor between RSV (+) and RSV (−) groups was CRP (*p*-value: 0.013).

Table 5 presents the ROC analysis results for the RSV (+) group according to age group. For children over two years old, the AUC was 0.947 (0.861–1.000), sensitivity was 92%, and specificity 100% for CRP, with a cutoff value of greater than one. Additionally, the NLR with a cutoff value of greater than 1.5 had 100% specificity and an AUC of 0.867 (0.710–1.000). However, for children under two years of age, the AUCs were 0.840 (95%CI: 0.773–0.908) for NLR with a cutoff value of lower than 0.85, 0.853 (0.797–0.909) for PLR with a cutoff value of lower than 73, 0.806 (0.730–0.881) for SII with a cutoff value of lower than 280, and 0.831 (95%CI: 0.731–0.931) for CRP with a cutoff value of greater than 1.5. Specificity values were over 80%.

## 4. Discussion

To the best of our knowledge, this is the first study to evaluate the effect of RSV infection on hematological indices such as NLR, PLR, and SII obtained from CBC analysis. Our study demonstrated that the NLR, PLR, and SII were lower in the RSV (+) group than in the RSV (−) group. Interestingly, several factors were significantly different for children under two years of age between the RSV (+) and (−) groups, whereas the only factor that differed significantly between the RSV (+) and (−) groups for children older than two years was CRP. Among all inflammatory parameters, CRP was found to have the highest sensitivity and specificity. High-grade inflammatory status appears to be a key component of RSV infection.

Various abnormalities can be observed in the hematological system in RSV infection. These include anemia, thrombocytopenia, thrombocytosis, leukopenia, and leukocytosis. In the current study, HGB and HCT were significantly lower in the RSV (+) group compared to the RSV (−) group, while lymphocyte and WBC counts were higher. Our results support the idea that RSV (+) patients are associated with a reduction in HCT and HGB, regardless of whether the infection is detected by rapid assay or culture. We propose that the factor causing acute bronchitis and bronchiolitis is related to viral infection, and therefore the lymphocyte count is high. However, no difference was found in neutrophil and platelet levels between groups. The results of our study showed that anemia, thrombocytopenia, leukocytosis, and lymphocytosis dominate in RSV patients. Unlike other viral infections, the immune response of the host, especially the direct cytotoxic effect of the virus, plays a role in the pathogenesis of RSV. It has been observed that antigenemia was very high in infant lung tissues examined after fatal RSV infections, but CD8-positive lymphocytes and natural killer cells were found to be low in number [11]. Again, in experimental studies in humans, it has been shown that the clinical course is directly proportional to the viral load. In addition, the lower cytokine response in RSV compared to influenza (flu) suggests that the virus has a direct cytotoxic effect in pathogenesis [12].

Saijo et al. [13] reported that the WBC and neutrophil counts in RSV lobar pneumonia cases were significantly greater than those in RSV bronchiolitis and bronchopneumonia cases. The probability of abnormal WBC counts of <5000 and 15,000–30,000 being associated with a concurrent serious bacterial infection was very low and no different from that of a normal WBC count in febrile patients admitted with RSV LRTI [14]. In a study in Beijing, 1860 patients were screened in a fever clinic [15]. The study enrolled 136 (7.31%) patients with positive nasal and pharyngeal swab tests for influenza A virus, influenza B virus, 2019-nCoV, or RSV [15]. COVID-19 patients had a lower WBC count and neutrophil count than the influenza A virus infection group and RSV infection group. There was no difference in lymphocyte count between the COVID-19 group and the RSV infection group. The RSV infection group had lower HGB levels than the other groups. Prozan et al. [7] compared hematological parameters in COVID-19 with two other common respiratory viruses, FLU (A and B) and RSV. Neutrophil counts were higher in patients with RSV compared to those with COVID-19 and flu virus. These differences between studies are probably related to virus type and load.

NLR, which is easily calculated from a routine blood test by dividing the absolute neutrophil count by the absolute lymphocyte count, is an easily available, inexpensive parameter that provides insight into the cellular immune and systemic inflammatory response. In addition, the inflammatory response can stimulate apoptosis in neutrophils and lymphocytes [16,17]. Recently, studies have reported that NLR is more reliable than neutrophil count or lymphocyte count alone in the prognosis of various diseases and in predicting patient survival. This immune system dysregulation can be used as a marker of disease activity caused by the virus [18]. In the case of a systemic inflammatory response, a decrease in lymphocyte count, an increase in neutrophil count, and a relative increase in NLR have been shown in previous studies [19]. In the current study, NLRs were significantly lower in the RSV (+) group compared to the RSV (−) group. According to the ROC analysis, NLR has the highest specificity (95.3%) out of the inflammatory indices (PLR and SII). The reason for the decrease in NLR is that lymphocyte counts were significantly higher in RSV (+) patients than in RSV (−) patients, and there was no difference in neutrophil count between the two groups. In a retrospective observational study at the Tel Aviv Sourasky Medical Center, NLR at admission was lower and had more prognostic value in COVID-19 patients when compared to flu and RSV [7]. The other reason for the difference between the NLR value in our study and the NLR in the study of Prozan et al. [7] is likely due to the small number of patients in our study. Following the neutrophilic response, RSV infection is associated with a pulmonary CD8 T-cell response, and lymphocyte levels rise, coinciding with viral clearance [20]. As seen in our study, theoretically, a low NLR would be expected to imply a favorable prognosis. Nonetheless, in our real-life, large cohort of RSV patients, NLR at admission did not have any prognostic value [7].

However, there are many factors that can affect platelet change in the clinic. In many clinical studies, platelets have been shown to play an important role in the hemostasis, inflammation, and immune processes. As a new type of inflammation index, PLR mainly reflects the level of systemic inflammation. In a study from China, the PLRs of patients with severe COVID-19 were significantly higher than those of non-severe patients. Nevertheless, the risks of WBC, CRP, PLR, and derived NLR ratio (d-NLR) were unclear. Elevated age and NLR can be considered independent biomarkers for indicating poor clinical outcomes [19]. NLR and PLR also had a certain degree of accuracy in the diagnosis of viral infections in children with influenza A [21]. The changes in the PLR in peripheral blood during treatment could reflect the disease progression and prognosis of COVID-19 patients. The PLR of patients means the degree of cytokine storm, which might provide a new indicator in the monitoring of patients with COVID-19 [22].

An innovative marker called the SII can predict the prognosis for other inflammatory diseases [20,21,22,23,24,25]. In previous research, it has been reported that CD8+T cells and IFN-γ have protective roles, while neutrophilic inflammation is incriminated as a harmful response. These may represent important therapeutic targets to modulate the immunopathogenesis of RSV infection [26]. In the current study, SIIs were significantly lower in the RSV (+) group compared to the RSV (−) group. The reason for the decrease in SII is that lymphocyte counts were significantly higher in patients with RSV (+) than those with RSV (−), and there was no difference in neutrophil and platelet count between groups. According to these results, it was concluded that the SII is a proinflammatory marker of systemic inflammation that can be effectively used to independently predict RSV. It may be an auxiliary parameter in the planning of the treatment process for RSV patients. The SII comprehensively summarizes the balance between the immunity and inflammatory status of the host. It has already been suggested as a prognostic biomarker in sepsis patients [22]. We believe that revealing the relationship between SII and RSV viral dynamics will contribute to the clinic.

CRP is one of the most commonly used inflammation markers for both acute and chronic inflammation. Serum CRP levels may increase during the course of various diseases, which indicates inflammation and tissue damage [27,28,29,30]. CRP levels are elevated in 75–93% of COVID-19 patients [27]. They increase in the early phase of the disease, and there is a positive correlation between rising CRP value and the severity of the disease [29]. In a study by Tan et al. [27], it was evaluated that CRP values above 20 mg/L may be an early marker for severe disease. Similarly, in the current study, the CRP values of patients with RSV were found to be significantly higher than those in the control group. CRP was found to have both high sensitivity (80.4%) and specificity (82.4%) among all parameters. High-grade inflammatory status appears to be a key component of infection with RSV. CRP is a fast and relatively easy method with high sensitivity and specificity, and it maintains its place in the rapid diagnosis of RSV. Papan et al. [30] evaluated the factors associated with antibiotic use in infections due to RSV and flu virus in young children. In their cohort, the rate of antibiotic utilization was comparable between RSV and flu patients, while for both groups, distinct clinical presentation and a high CRP value were associated with higher antibiotic use. Higdon et al. [31] evaluated the sensitivity and specificity of CRP for identifying bacterial vs. RSV pneumonia in the Pneumonia Etiology Research for Child Health (PERCH) multicenter case–control study. While CRP had imperfect specificity for distinguishing bacterial from RSV pneumonia and therefore limited use as a diagnostic tool, the clear association of elevated CRP with bacterial pneumonia makes it potentially useful in epidemiologic studies of bacterial pneumonia, as cases with low CRP could be assumed to have a lower probability of bacterial etiology than cases with high CRP. The sample in our study mostly consists of patients under two years old. Age can be an important confounder in pediatric groups. CRP is also a suitable marker for patients over two years old. According to our results, we recommend that when patients present with LRTI have high CRP in their laboratory results and are younger than two years old, RSV should be considered first among possible viral agents and evaluated with a rapid antigen test. However, for those under two years of age, NLR, PLR, and SII markers perform as well as CRP. Thus, if the causative agent of the disease is known, disease management is facilitated, and unnecessary antibiotics are avoided.

### Limitations of the Study

Our study is a single-center and retrospective study. It was conducted on a limited group of cases. On the other hand, although hematological indices are a biomarker that has been studied in many diseases and case groups, reference values have not yet been determined. The majority of our sample consists of patients under two years old. Further investigations, including into other viral diseases, will be required to verify these results for RSV.

Our study presents the conclusion that CRP values perform better as markers than other blood parameters. Especially in children over two years of age with RSV, high CRP levels can be a helpful parameter in the planning of the treatment process for patients diagnosed with pneumonia and LRTI in the emergency department. However, NLR, PLR, and SII values are additional parameters that can still give clinical insight alongside other parameters. New-generation inflammation markers, which are simple, inexpensive, easily accessible and non-invasive, obtained by using CBC parameters, are advantageous to use. Our study may be an inspiration for investigating the effect of other inflammatory parameters, such as the hematological index, on RSV infection. RSV infection is a health problem and may be associated with asthma as the patient ages. At the same time, it is still not possible to give a clear lower and upper limit for the new-generation inflammation markers used in our study in healthy children. Therefore, there is a need for long-term studies that include long-term observation of patients with RSV. We believe that our study will be a pioneer in this regard, as it is the first study to examine new-generation inflammation markers in RSV patients.

## Figures and Tables

**Figure 1 viruses-15-01245-f001:**
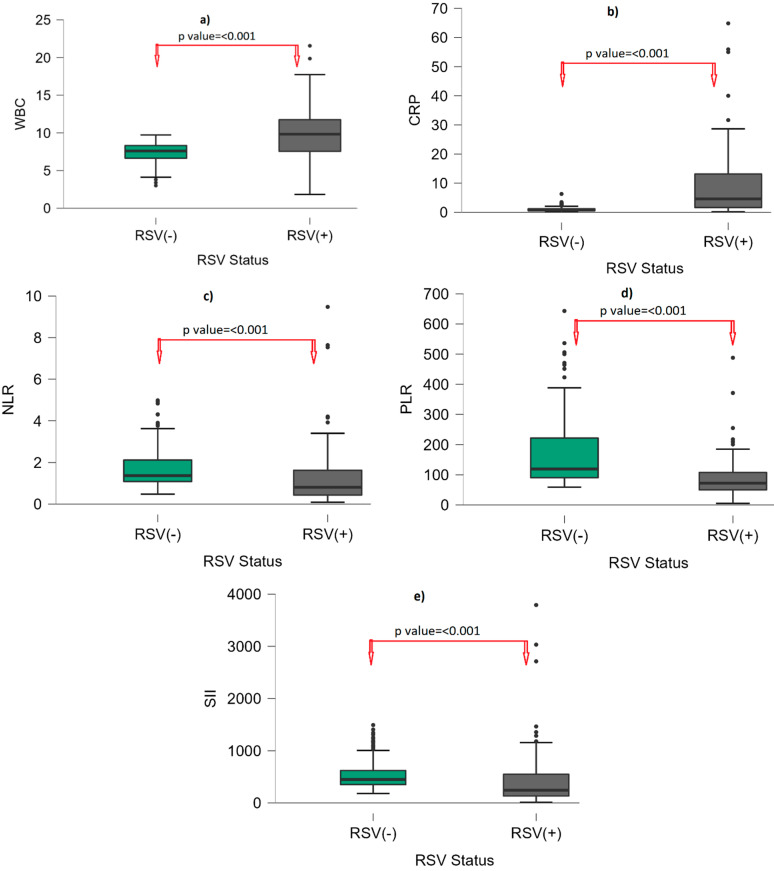
Box-plot graph by RSV status. (**a**): WBC; (**b**): CRP; (**c**): NLR; (**d**): PLR; (**e**): SII. Points (●) represent outliers. Abbreviations: WBC: White blood cell; CRP: C-reactive protein; NLR: neutrophil–to–lymphocyte ratio; PLR: platelet–to–lymphocyte ratio; SII: systemic immune–inflammatory index; RSV: respiratory syncytial virus.

**Figure 2 viruses-15-01245-f002:**
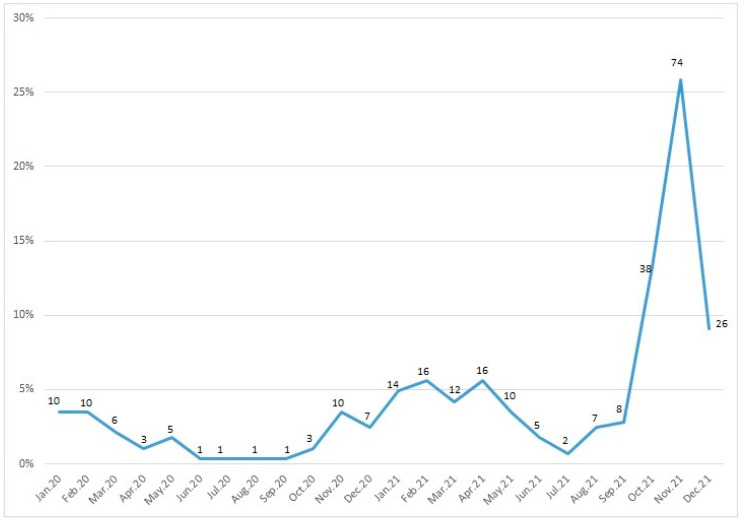
Seasonal distribution of respiratory syncytial virus. Note: Data labels show patient numbers, y-axis show percentage.

**Figure 3 viruses-15-01245-f003:**
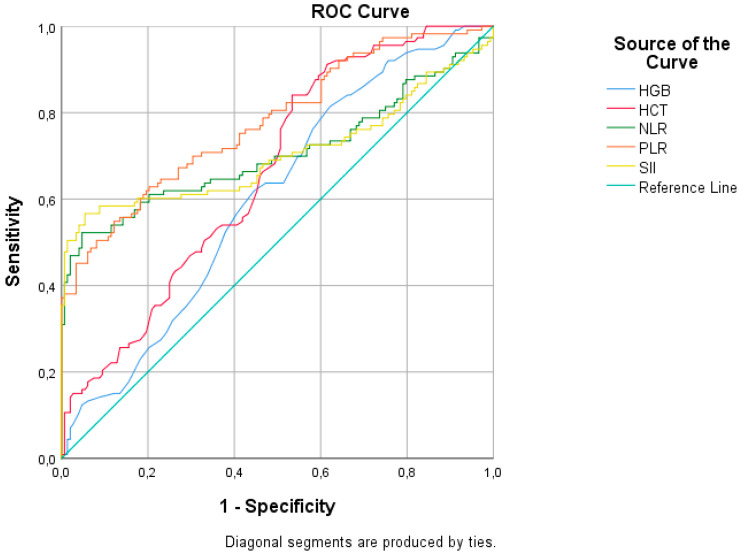
ROC analysis of hematologic parameters for respiratory syncytial virus infection.

**Table 1 viruses-15-01245-t001:** Demographics and Laboratory findings of respiratory syncytial virus (RSV)-infected and non-infected groups.

	RSV (−) (n: 148; 51.7%)	RSV (+) (n: 138; 48.3%)	*p*-Value
Gender			
Male	87 (58.8%)	73 (52.9%)	0.316 ^†^
Female	61 (41.2%)	65 (47.1%)	
Age(months)	8 (5–11); 8.6 ± 8.9	13 (5–31.3); 23.2 ± 25.2	**<0.001 ***
≤24	145 (98.0%)	92 (66.7%)	
25–59	2 (1.4%)	30 (21.7%)	**<0.001 ^†^**
≥60	1 (0.7%)	16 (11.6%)	
WBC (×10^6^/µL)	7.6 (6.61–8.39)	9.83 (7.53–11.83)	**<0.001 ***
HGB (g/dL)	12 (11.1–12.88)	11.6 (10.8–12.2)	**0.004 ***
HCT (%)	35.5 (33.15–38.78)	34.1 (32.1–35.7)	**<0.001 ***
PLT (×10^3^/mL)	324.4 (292.5–357)	317 (256–396.5)	0.723 *
Lymphocyte (10^3^/µL)	2.78 (1.4–3.6)	4.34 (2.65–6.36)	**<0.001 ***
Neutrophil (10^3^/µL)	3.83 (2.85–4.49)	3.51 (2.26–5.85)	0.999 *
Monocyte (10^3^/µL)	0.9 (0.65–1.19)	0.92 (0.71–1.24)	0.444 *
CRP (mg/L)	0.72 (0.55–1.25)	4.59 (1.57–13.63)	**<0.001 ***
NLR	1.37 (1.08–2.16)	0.81 (0.41–1.64)	**<0.001 ***
PLR	119.31 (90.32–223.07)	72.17 (49.63–108.26)	**<0.001 ***
SII	449.5 (349.25–622)	243.40 (127.03–554.42)	**<0.001 ***

Abbreviations: WBC: White blood cell; HGB: hemoglobin; HCT: hematocrit; PLT: platelet; CRP: C-reactive protein; NLR: neutrophil–to–lymphocyte ratio; PLR: platelet–to–lymphocyte ratio; SII: systemic immune–inflammatory index. Categorical variables are presented as n (%); continuous variables are presented as median (IQR). *: Mann–Whitney U test; ^†^: Chi-Square test was applied.

**Table 2 viruses-15-01245-t002:** Comparison of respiratory syncytial virus-positive and -negative patients according to their clinical findings.

	RSV (−) (n: 148; 51.7%)	RSV (+) (n: 138; 48.3%)	*p*
Fever	111 (75%)	138 (100%)	0.001
Cough	127 (85.8%)	138 (100%)	0.001
Vomiting	39 (26.3%)	20 (14.5%)	0.013
Wheezing	148 (100%)	138 (100%)	-
Apnea	15 (10.1%)	9 (6.5%)	0.271
Cyanosis	53 (35.8%)	55 (40%)	0.481
Groan	143 (96.6%)	136 (98.5%)	0.291
Acute otitis media	24 (16.2%)	40 (29.0%)	0.010
Rhinorrhea	37 (25.0%)	31 (22.5%)	0.615
Dyspnea	83 (56%)	94 (68%)	0.036
Infiltration on chest X-ray	84 (56.7%)	40 (29.0%)	0.001
Hospitalization	-	16 (11.6%)	-
Length hospitalization	-	1 (3–9); (Min: 2; Max: 20)	-

**Table 3 viruses-15-01245-t003:** ROC analysis for patients with respiratory syncytial virus.

	AUC	CI 95%	*p*-Value	Cutoff	Sensitivity	Specificity
WBC	0.841	0.765, 0.917	<0.001	9 ^a^	63.7%	**87.8%**
LYM	0.703	0.618, 0.788	<0.001	4.1 ^a^	57.5%	**87.8%**
HGB	0.604	0.536, 0.672	0.004	12.5 ^b^	**84.1%**	33.1%
HCT	0.661	0.596, 0.726	<0.001	37.5 ^b^	**92.0%**	36.5%
NLR	0.706	0.636, 0.776	<0.001	0.85 ^b^	51.3%	**95.3%**
PLR	0.779	0.722, 0.836	<0.001	73 ^b^	50.4%	**90.5%**
SII	0.705	0.633, 0.776	<0.001	280 ^b^	56.6%	**93.9%**
CRP	0.869	0.800, 0.937	<0.001	1.5 ^a^	**80.4%**	**82.4%**

Abbreviations: WBC: White blood cell; HGB: hemoglobin; HCT: hematocrit; LYM: lymphocytes; CRP: C-reactive protein; NLR: neutrophil–to–lymphocyte ratio; PLR: platelet–to–lymphocyte ratio; SII: systemic immune–inflammatory index. ^a^: Larger values; ^b^: Smaller values indicates RSV (+).

**Table 4 viruses-15-01245-t004:** Laboratory findings of respiratory syncytial virus-infected and non-infected patients by age group.

	Age Group (Years)					
	≤2 (n: 237; %82.9)			>2 (n: 49; %17.1)		
	RSV (−) (n: 145; %61.2)	RSV (+) (n: 92; %38.8)		RSV (−) (n: 3; %6.1)	RSV (+) (n: 46; %93.9)	
	Median (25–75p)	Median (25–75p)	*p*	Median (25–75p)	Median (25–75p)	*p*
WBC (×10^6^/µL)	7.6 (6.65–8.42)	10.38 (8.56–12.1)	<0.001	7.32 (6.19–7.8)	8.6 (6.81–10.72)	0.342
HGB (g/dL)	12 (11.1–12.8)	11.4 (10.65–12)	<0.001	12.2 (11.5–13.9)	12.2 (11.6–12.5)	0.719
HCT (%)	35.5 (33.1–38.7)	33.2 (31.3–35.35)	<0.001	36.9 (33.9–39.6)	35.1 (33.8–36.9)	0.441
PLT (×10^3^/mL)	324.2 (292–354)	349 (271.5–420.5)	0.089	358 (298–368)	289 (238–339)	0.111
Lymphocyte (10^3^/µL)	2.7 (1.4–3.6)	5.5 (4.265–7.387)	<0.001	3.47 (2.88–3.74)	2.49 (2.077–3.19)	0.158
Neutrophil (10^3^/µL)	3.82 (2.88–4.43)	2.86 (2.1–4.63)	0.061	4.52 (2.61–4.75)	4.73 (2.97–6.74)	0.456
Monocyte (10^3^/µL)	0.9 (0.67–1.19)	0.995 (0.8–1.29)	0.066	0.89 (0.58–0.9)	0.84 (0.63–1.03)	0.98
CRP (mg/L)	0.74 (0.55–1.26)	2.78 (1.49–12.92)	<0.001	0.62 (0.6–0.7)	7.93 (3.96–13.86)	0.013
NLR	1.39 (1.09–2.18)	0.56 (0.31–0.99)	<0.001	1.21 (0.91–1.37)	1.63 (1.17–2.34)	0.19
PLR	119.33 (90.51–223.53)	57.82 (46.74–85.94)	<0.001	98.4 (85.88–124.31)	102.66 (77.58–143.98)	0.939
SII	455 (353–623)	185.08 (95.86–384)	<0.001	408 (324–445)	460.74 (275.05–774.62)	0.488

Abbreviations: WBC: White blood cell; HGB: hemoglobin; HCT: hematocrit; PLT: platelet; CRP: C-reactive protein; NLR: neutrophil–to–lymphocyte ratio; PLR: platelet–to–lymphocyte ratio; SII: systemic immune–inflammatory index.

**Table 5 viruses-15-01245-t005:** ROC analysis for patients with respiratory syncytial virus according to age group.

	Age Group (Years)											
	≤2 (n: 237; %82.9)						>2 (n: 49; %17.1)					
	AUC	CI 95%	*p*	Cutoff	Sensitivity	Specificity	AUC	CI 95%	*p*	Cutoff	Sensitivity	Specificity
WBC	0.915	0.843, 0.986	<0.001	9 ^a^	72.4%	87.6%	0.787	0.620, 0.953	0.11	-	-	-
LYM	0.852	0.765, 0.940	<0.001	4.1 ^a^	77.6%	87.6%	0.72	0.540, 0.900	0.22	-	-	-
HGB	0.669	0.598, 0.741	<0.001	12.5 ^b^	89.5%	33.1%	0.593	0.273, 0.913	0.603	-	-	-
HCT	0.720	0.653, 0.787	<0.001	37 ^b^	93.4%	41.4%	0.633	0.337, 0.930	0.458	-	-	-
NLR	0.840	0.773, 0.908	<0.001	0.85 ^b^	71.1%	95.2%	0.867	0.710, 1.000	0.041	1.5 ^a^	54.1%	100%
PLR	0.853	0.797, 0.909	<0.001	73 ^b^	64.5%	90.3%	0.52	0.280, 0.760	0.911	-	-	-
SII	0.806	0.730, 0.881	<0.001	280 ^b^	71.1%	93.8%	0.773	0.603, 0.944	0.128	-	-	-
CRP	0.831	0.731, 0.931	<0.001	1.5 ^a^	74.2%	82.1%	0.947	0.861, 1.000	0.013	1 ^a^	92%	100%
				1.3 ^a^	80.6%	75.9%						
				2.1 ^a^	64.5%	91.7%						

Abbreviations: WBC: White blood cell; HGB: hemoglobin; HCT: hematocrit; PLT: platelet; LYM: lymphocyte; CRP: C-reactive protein; NLR: neutrophil–to–lymphocyte ratio; PLR: platelet–to–lymphocyte ratio; SII: systemic immune–inflammatory index. ^a^: Larger values; ^b^: Smaller values indicates RSV (+).

## Data Availability

The data underlying this article are available in the article. If needed, please contact the corresponding author. The email address is dmemhs@gmail.com.

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
