# Peer review of "New Generation of Systemic Inflammatory Markers for Respiratory Syncytial Virus Infection in Children"

_viruses, 2023, doi:10.3390/v15061245_

Round 1

Reviewer 1 Report (New Reviewer)

1. How did the authors define RSV (-)ve group? It is apparently healthy or other febrile illness patients? Did the authors include Influenza and COVID-19 as the authors make diagnosis uding multiplex PCR. The readers are not clearly understanding the above issue. Please clearly rewrite it.

2. At Figure-1, all X -axis were not found. Please revise it.

3.At Table-3, the authors showed sensitivity and specificity of each indicators according to the age group (less than and older than 2 years).  Why did the authors not describe for some indictors for more than 2 years old group? Please add and describe it. Regarding the alignment of the data, we usually describe as right alignment as it can easily if there is no specific instructions from journal. Left alignment is difficult to see. Moreover, for description of 95% CI, we usually not express as “-“. Please use “’,”. (eg 95% CI 12.3, 14.1)

4. Regarding the discussion, the authors make discussion on SII and other predictors But the authors discussed a lot about the nature of the this indicators such as SII and COVID-19 and malignancy. Those are not associated with this study. Please remove those parts and discussed on your data.

5. The authors noted that there is difference among the indictors such as Hb and HCT. What is the mechanism and please discuss it.

6. Regarding your biomarkers, the reviewer noted that the significant was found mostly under 2 years age group. Not significant association more than 2 years old group. So those indicators can be used only under two years old patients? Please make discussion at the revised one. The sample size is also totally different between under two years and more than 2.

Author Response

Dear Editor,

First, we would like thank the reviewers for the helpful comments, which led us to conduct appropriate experiments. The revised manuscript has subsequently been rewritten to address these concerns and comments of the reviewers.

We are grateful for your understanding and cooperation in this matter.

We believe that the manuscript is now suitable for review. We look forward to your reply.

RESPONSE TO REVIEWERS:

Reviewer 1

Comments and Suggestions for Authors

  1. How did the authors define RSV (-)ve group? It is apparently healthy or other febrile illness patients? Did the authors include Influenza and COVID-19 as the authors make diagnosis using multiplex PCR. The readers are not clearly understanding the above issue. Please clearly rewrite it.

Patients were grouped as positive and negative according to RSV results. A total of 286 consecutive patients, 138 RSV (+) diagnosed with LRTI and 148 RSV (-) with community-acquired pneumonia (CAP) in nasopharyngeal swab samples, between the ages of 0-12 and admitted to the Medicine Hospital (pediatric outpatient clinic and pediatric emergency clinic) between January 1, 2020 and January 1, 2022 were included in the study.

First RSV infection is determined by rapid antigen test. After RSV were confirmed by multiplex PCR kit (INFLUENZA A/B, SARS-CoV-2, RSV). COVID-19 and influenza patients were not included in the study because the number of patients with COVID-19 was insufficient. 

  1. At Figure-1, all X -axis were not found. Please revise it.

It is corrected.

3.At Table-3, the authors showed sensitivity and specificity of each indicators according to the age group (less than and older than 2 years).  Why did the authors not describe for some indictors for more than 2 years old group? Please add and describe it. Regarding the alignment of the data, we usually describe as right alignment as it can easily if there is no specific instructions from journal. Left alignment is difficult to see. Moreover, for description of 95% CI, we usually not express as “-“. Please use “’,”. (eg 95% CI 12.3, 14.1)

We did not describe them because the ROC analysis was not statistically significant. It would be misinterpretation to give a cut-off here.

In addition, We have corrected the alignment and CI expressions in tables.

  1. Regarding the discussion, the authors make discussion on SII and other predictors But the authors discussed a lot about the nature of the this indicators such as SII and COVID-19 and malignancy. Those are not associated with this study. Please remove those parts and discussed on your data.

Data on SII, COVID-19 and malignancy were removed from the discussion.

  1. The authors noted that there is difference among the indictors such as Hb and HCT. What is the mechanism and please discuss it.

The indictors such as Hb and HCT were added to the discussion.

  1. Regarding your biomarkers, the reviewer noted that the significant was found mostly under 2 years age group. Not significant association more than 2 years old group. So those indicators can be used only under two years old patients? Please make discussion at the revised one. The sample size is also totally different between under two years and more than 2.

Yes, we are aware of this result. The sample of our study mostly consists of the group under 2 years old. Age can be an important confounder in the pediatric group. Therefore, we wanted to make a subgroup analysis for age. CRP is also a suitable marker over 2 years of age. However, under 2 years of age, NLR, PLR, SII markers as well as CRP perform well. A limitation of our study is that the majority of our sample consisted of the under-2-year-old group.

Reviewer 2 Report (New Reviewer)

This study is evaluating the relationship between various biomarkers with clinical findings of RSV infection among children with the diagnosis of lower respiratory tract infection. My comment are...

In Abstract:

It is written: "This retrospective study included 286 consecutive patients between 0-12 years of age with 138 RSV (+) (48.25%) 35 and age-matched 148 RSV (-) (51.75%).". Please elaborate clearly which patients were included consecutivetly? And if the RSV- patients were age matched to those RSV+, why the number is not even?

What was the etiology of LRTI in RSV- patients? Viral or bacterial or unknown?

It is written "The AUC was statistically significant for parameters..." but it is unclear in which patients. All of them?

Please write clearly the conclusion: "Thus, if the cause of the disease is known, it is easier in terms of disease management and unnecessary antibiotics are avoided." with refferal to the findings of the study.

In Introduction:

I suggest to incorporate the reference: https://pubmed.ncbi.nlm.nih.gov/35598608/

Please elaborate more the reason for assuming that CRP, NLR, PLR and SII can be used as a pre-test in RSV. And also elaborate the meaning "pre-test".

In Material and Method: 

In line 86, age matched is not mentioned. Please elaborate more precisely the patient enrollment.

Please explain the meaning of "COVID-19 and influenza patients were not included in the study because the number of patients with COVID-19 was insufficient." In which way was the bacterial infection excluded? What is assumed as the etiology of LRTI in RSV- patients?

Please cite the refence based on which LRTI was defined.

Inclusion criteria is mentioning only RSV+ as inclusion criterian. But also RSV- were included and analyzed in the manuscript. Please define inclusion criteraia as precise as possible. Why is the cutoff set at 12 years?

In line 117, what is the meaning "recurrent RSV infections". Does this mean that RSV+ who had RSV infection before were excluded? Please state the number how many of those were found and twhat was the reason to exclude them from this study?

When was an analysis of C-reactive protein done? At admission? Was there a protocol for CRP analysis i.e. 1st day of the febrile disease, 2nd day...?

In Results:

Line 36 is stating "age-matched", but in Table 1, there is a significant difference regarding the age between RSV+ and RSV- groups. Please explain.

Please prapare the Tables according to the rules, explaining all abbreviations.

It is stated "HGB and HCT were statistically significantly lower in RSV (+) than RSV (-) (p:0.012; 144 p<0.001, respectively)." Please discuss this in the Discussion. Was this expected and if yes, why?

Why T-IgE was tested in out patient and emergeny clinic? Why only in RSV+ patients? 

Among RSV- patients 84 (56.7%) had infiltration on chesst x-ray and 53 (35.8%) were cyanotic. However, nobody was hospitalized? How were they treated?

Acute otitis media was diagnosed in 24 (15.9%) RSV- and 40 (28.8%) RSV+ patients. Infection of the middle ear can be viral, bacterial, or coinfection. Was bacterail infection definetely excluded in all of them?

Infiltration was seen in 40 RSV + patients. Please explain how the percentage of 61,7% was calculated. Please check all other percentages, as well.

In Fig 2 please, beside percentages, indicate the numebr of patients and also present all months within the studied period and add year next to the month.

In Table 4, please add the number how many RSV+/- were in each subgroup and also revise the text explaining this Table since I find it a little confusing.

In Discussion:

Please explain how in children over 2 years of age with RSV, high CRP levels can be a helpful parameter in the planning of the treatment process of patients diagnosed with pneumonia and LRTI in the emergency department.

The native English speaker should check the revis eversion before resubmitting it.

Author Response

Dear Editor,

First, we would like thank the reviewers for the helpful comments, which led us to conduct appropriate experiments. The revised manuscript has subsequently been rewritten to address these concerns and comments of the reviewers.

We are grateful for your understanding and cooperation in this matter.

We believe that the manuscript is now suitable for review. We look forward to your reply.

RESPONSE TO REVIEWERS:

Reviewer 2

Comments and Suggestions for Authors

This study is evaluating the relationship between various biomarkers with clinical findings of RSV infection among children with the diagnosis of lower respiratory tract infection. My comment are...

In Abstract:

It is written: "This retrospective study included 286 consecutive patients between 0-12 years of age with 138 RSV (+) (48.25%) 35 and age-matched 148 RSV (-) (51.75%).". Please elaborate clearly which patients were included consecutivetly? And if the RSV- patients were age matched to those RSV+, why the number is not even?

Patients were grouped as positive and negative according to RSV results. A total of 286 consecutive patients, 138 RSV (+) diagnosed with LRTI and 148 RSV (-) with community-acquired pneumonia (CAP) in nasopharyngeal swab samples, between the ages of 0-12 and admitted to the Medicine Hospital (pediatric outpatient clinic and pediatric emergency clinic) between January 1, 2020 and January 1, 2022 were included in the study

What was the etiology of LRTI in RSV- patients? Viral or bacterial or unknown?

Patients were grouped as positive and negative according to RSV results. A total of 286 consecutive patients, 138 RSV (+) diagnosed with LRTI and 148 RSV (-) with community-acquired pneumonia (CAP) in nasopharyngeal swab samples, between the ages of 0-12 and admitted to the Medicine Hospital (pediatric outpatient clinic and pediatric emergency clinic) between January 1, 2020 and January 1, 2022 were included in the study.

It is written "The AUC was statistically significant for parameters..." but it is unclear in which patients. All of them?

It is corrected and corrections are seen as red color in the text.

The AUC was statistically significant for parameters in all groups;……… While ROC analysis results gave similar results for under 2 years old, only CRP and NLR were statistically significant over 2 years old.

Please write clearly the conclusion: "Thus, if the cause of the disease is known, it is easier in terms of disease management and unnecessary antibiotics are avoided." with refferal to the findings of the study.

In Introduction:

I suggest to incorporate the reference: https://pubmed.ncbi.nlm.nih.gov/35598608/

Please elaborate more the reason for assuming that CRP, NLR, PLR and SII can be used as a pre-test in RSV. And also elaborate the meaning "pre-test".

Pretest added to the introduction section.

In Material and Method:

In line 86, age matched is not mentioned. Please elaborate more precisely the patient enrollment.

We thank to reviewer for their attention. We didn't do age-matcing. This is incorrect usage. We removed it.

Please explain the meaning of "COVID-19 and influenza patients were not included in the study because the number of patients with COVID-19 was insufficient." In which way was the bacterial infection excluded?

In the retrospective study, it was observed that all RSV-positive patients received antibiotic treatment. We can say that the reason for this is the inability to distinguish between viral and bacterial infections at the time of admission and the fear of developing secondary infection. Detection of the causative agent is important in terms of providing appropriate isolation conditions and preventing unnecessary antibiotic use.

What is assumed as the etiology of LRTI in RSV- patients?

Patients were grouped as positive and negative according to RSV results. A total of 286 consecutive patients, 138 RSV (+) diagnosed with LRTI and 148 RSV (-) with community-acquired pneumonia (CAP) in nasopharyngeal swab samples, between the ages of 0-12 and admitted to the Medicine Hospital (pediatric outpatient clinic and pediatric emergency clinic) between January 1, 2020 and January 1, 2022 were included in the study. In CAP, the most common bacterial agent is Streptococcus pneumoniae.

Please cite the refence based on which LRTI was defined.

It was added to the references as literature number 10.

Inclusion criteria is mentioning only RSV+ as inclusion criterian. But also RSV- were included and analyzed in the manuscript. Please define inclusion criteraia as precise as possible.

Necessary additions were made according to your suggestions.

Why is the cutoff set at 12 years?

Since the number of patients over the age of 12 was low (n=5), they were not included in the study.

In line 117, what is the meaning "recurrent RSV infections". Does this mean that RSV+ who had RSV infection before were excluded? Please state the number how many of those were found and twhat was the reason to exclude them from this study?

Recurrent patients were excluded from the study due to the small number  (n=6) of patients.

was an When analysis of C-reactive protein done? At admission? Was there a protocol for CRP analysis i.e. 1st day of the febrile disease, 2nd day...?

CRP analysis was done at the admission. There is no protocol for CRP analysis.

In Results:

Line 36 is stating "age-matched", but in Table 1, there is a significant difference regarding the age between RSV+ and RSV- groups. Please explain.

We thank to reviewer for their attention. We didn't do age-matcing. This is incorrect usage. We removed it.

Please prapare the Tables according to the rules, explaining all abbreviations.

It is corrected.

It is stated "HGB and HCT were statistically significantly lower in RSV (+) than RSV (-) (p:0.012; 144 p<0.001, respectively)." Please discuss this in the Discussion. Was this expected and if yes, why?

Necessary additions were made according to your suggestions.

Why T-IgE was tested in out patient and emergeny clinic? Why only in RSV+ patients?

IgE-mediated hypersensitivity: After bronchiolitis caused by RSV, recurrent wheezing attacks are seen in 22-76%. The relationship between RSV infection and wheezing has been investigated in some patients because RSV-infected cells are coated with IgE. While RSV specific IgE was detected in the nasopharyngeal secretions of babies with wheezing at a rate of 45%, it was not detected at all in those without wheezing. In addition, the peak RSV titer was correlated with the severity of hypoxia, which is an objective indicator of the severity of the disease (A, B).

  1. La Via W.V, Marks M.I, Stutman H.R.: Respiratory syncytial virus puzzle: Clinical features, pathophysiology, treatment and prevention. J Pediatr 1992; 121:4, 503-510.
  2. .Jalowayski AA, Walpita P, Puryear BA, Connor JD. Rapid detection of respiratory

syncytial virus in nasopharyngeal specimens obtained with rhinoprobe scraper. J Clin

Microbiol, 1990;28:738-739.

It was requested in children with allergic reactions. However, we have removed the IgE results from the table. Since there were only 44 patients, we excluded it from the table.

Among RSV- patients 84 (56.7%) had infiltration on chesst x-ray and 53 (35.8%) were cyanotic. However, nobody was hospitalized? How were they treated?

Amoxicillin is the treatment of choice for Streptococcus pneumoniae, and azithromycin is preferred for Mycoplasma pneumoniae.

Acute otitis media was diagnosed in 24 (15.9%) RSV- and 40 (28.8%) RSV+ patients. Infection of the middle ear can be viral, bacterial, or coinfection. Was bacterail infection definetely excluded in all of them?

Patients with CRP <5 were excluded.

Infiltration was seen in 40 RSV + patients. Please explain how the percentage of 61,7% was calculated. Please check all other percentages, as well.

In Fig 2 please, beside percentages, indicate the numebr of patients and also present all months within the studied period and add year next to the month.

Necessary arrangements have been made.

In Table 4, please add the number how many RSV+/- were in each subgroup and also revise the text explaining this Table since I find it a little confusing.

It is corrected. If the you consider it unnecessary, we can exclude the analyzes in age subgroups.

In Discussion:

Please explain how in children over 2 years of age with RSV, high CRP levels can be a helpful parameter in the planning of the treatment process of patients diagnosed with pneumonia and LRTI in the emergency department.

Necessary arrangements have been made.

Comments on the Quality of English Language

The native English speaker should check the revis eversion before resubmitting it.

The article has been re-edited.

Reviewer 3 Report (New Reviewer)

Please read the attachment. Thank you. 

Author Response

Dear Editor,

First, we would like thank the reviewers for the helpful comments, which led us to conduct appropriate experiments. The revised manuscript has subsequently been rewritten to address these concerns and comments of the reviewers.

We are grateful for your understanding and cooperation in this matter.

We believe that the manuscript is now suitable for review. We look forward to your reply.

RESPONSE TO REVIEWERS:

Reviewer 3

The systemic immune-inflammatory index (SII), neutrophil-to-lymphocyte ratio

(NLR), and platelet-to-lymphocyte ratio (PLR) were studied in children with LRTI

and respiratory syncytial virus (RSV) infection. Methods: The pediatric clinic

experiment ran from 2020–2022. Two hundred eighty-six consecutive patients

aged 0-12 with 138 RSV (+) (48.25%) and age-matched 148 (-) (51.75%) were

analyzed retrospectively. Chromatographic immunoassay on nasopharyngeal

swabs identified RSV antigen. Results: RSV (+) children had greater CRP, lower

NLR, PLR, and SII than RSV (-) children. RSV+ groups had 100% fever, cough, and

wheezing. RSV infections peaked in November, October, and December.

Leukocytes, CRP, NLR, PLR, and SII, had statistically significant AUCs. CRP is

sensitive (80.4%) and specific (82.4%) across all measurements. Conclusion: CRP

outperforms other blood measurements. RSV (+) LRTI patients had lower NLR,

PLR, and SII indexes than RSV (-) patients, indicating higher inflammation.

Knowing the cause simplifies disease treatment and reduces antibiotic use. In

conclusion, the reviewer found that the article has merits and could be acceptable

to publish in future forms. Therefore, please revise the manuscript according to

the reviewer's comments.

Thank you for your constructive comments.

  • The manuscript sounds well-structured and informative. The study aimed

to investigate the relationship between systemic immune-inflammatory

parameters (SII, NLR, and PLR) and clinical findings of RSV infection in

children with LRTI. The study included a reasonable sample size of 286

patients, which is a strength of the study.

  • The methods section clearly describes the study design and inclusion

criteria. However, no information is provided on the exclusion criteria, an

essential aspect of any study. It would be helpful to know whether any

patients were excluded and the reasons for their exclusion.

  • The results section presents the findings of the study clearly and concisely.

The authors report that CRP was significantly higher in patients with RSV

(+) than in those with RSV (-), while NLR, PLR, and SII were substantially

lower in RSV (+) patients. The authors also report the AUCs for each

immune-inflammatory parameter, which provides valuable information on

their diagnostic accuracy. However, the manuscript would benefit from

including confidence intervals for the AUCs.

Thank you for your constructive comments.

  • The discussion section provides a reasonable interpretation of the results

and places them in the context of previous research. The authors suggest

that CRP is superior to other immune-inflammatory parameters in

diagnosing RSV infection, which is a valuable finding. The authors also

suggest that knowing the cause of the disease can help with disease

management and avoid unnecessary antibiotics, which is an important

clinical implication.

Thank you for your constructive comments.

  • Title: the title is too long. Please shorten the manuscript title to no more than

ten words; the long ones will not be preferred to cite in the future.

Nowadays, many journals will charge the paper length pages, so

researchers intend to remove the manuscripts with long titles to reduce the

page length and APC charge.

The title has been shortened.

  • With every section, please mark it with a number. Please refer to the journal

template.

  • Please format the manuscript following the guidelines of the journal

template.

  • Table 1: Please revise the decimal number in the column of the p-value. A

mess of dots (.) and commas (,) exist. Also, please check the other tables.

  • Figure 1: the figure and its title should be on the same page.
  • Figure 2: Please add the title for both axes.
  • After tables, please use a dot (.) instead of (:). Please revise Table 1, Table 3,

Table 4, Table 5,

  • After figures, please use a dot (.) instead of (:). Please revise Figure 2,

Figure 3

  • Figure 3 is too small.

Necessary arrangements have been made.

  • How was the study conducted, and what were the inclusion criteria for

participants?

The criteria for inclusion and exclusion were rearranged.

  • What were the main findings of the study?

The patients presenting with LRTI have high CRP in their laboratory values and that the patient is younger than 2 years old, that RSV should be considered first among viral agents and evaluated with rapid antigen test. However, under 2 years of age, NLR, PLR, SII markers as well as CRP perform well.

  • How do the study’s findings compare to previous research on the relationship between immune-inflammatory parameters and RSV infection?

In previous studies, it has been reported that neutrophilic inflammation is incriminated as a harmful response, whereas CD8+ T cells and IFN-γ have protective roles. These may represent important therapeutic targets to modulate the immunopathogenesis of RSV infection.

  • According to the authors, which immune-inflammatory parameter is superior in diagnosing RSV infection?

The patients presenting with LRTI have high CRP in their laboratory values and that the patient is younger than 2 years old, that RSV should be considered first among viral agents and evaluated with rapid antigen test. However, under 2 years of age, NLR, PLR, SII markers as well as CRP perform well.

  • What are the clinical implications of the study's findings?

NLR, PLR and SII values are additional parameters that have the value of giving clinical insight by helping other parameters.

  • Were any patients excluded from the study, and if so, what were the exclusion criteria?

The criteria for inclusion and exclusion were rearranged.

  • How could the study be improved or expanded upon in future research?

Our work will guide the study of other immune-inflammatory parameter to be studied in RSV in the future. But the experiment design needs to include other viral infections as control. How would CRP, NLR, PLR and SII values be used to differentiate RSV v.s. COVID?

RSV is a major cause of morbidity in early childhood, in patients with cardiopulmonary disease, in immunocompromised individuals, and in the elderly. Therefore, it is necessary to evaluate these parameters in every age group in RSV(+) patients.

  • What is RSV, and why is it important to study its relationship with immune-inflammatory parameters?

RSV is a major cause of lower respiratory tract infections (such as bronchiolitis and pneumonia) (LRTI), especially in young infants.

NLR, PLR and SII, which are considered as new markers in systemic inflammatory disorders, reflect the immune response resulting from various stress stimuli.

  • What are some potential limitations of the study, and how might they affect the generalizability of the findings?

Our work will guide the study of other immune-inflammatory parameter to be studied in RSV in the future. But the experiment design needs to include other viral infections as control. How would CRP, NLR, PLR and SII values be used to differentiate RSV v.s. COVID?

RSV is a major cause of morbidity in early childhood, in patients with cardiopulmonary disease, in immunocompromised individuals, and in the elderly. Therefore, it is necessary to evaluate these parameters in every age group in RSV(+) patients.

Round 2

Reviewer 2 Report (New Reviewer)

I thank the authors for rewriting the manuscript and for changes they introduced according to my previous comments. However, there are still some issues that need to be improved like more precise inclusion criteria, methods etc.

English native speaker should check the manuscript once again due.

Author Response

Dear Editor,

First, we would like thank the reviewers for the helpful comments, which led us to conduct appropriate experiments. The revised manuscript has subsequently been rewritten to address these concerns and comments of the reviewers. English language edited by MDPI.

We are grateful for your understanding and cooperation in this matter.

We believe that the manuscript is now suitable for review. We look forward to your reply.

RESPONSE TO REVIEWERS:

Reviewer 2

Comments and Suggestions for Authors

I thank the authors for rewriting the manuscript and for changes they introduced according to my previous comments. However, there are still some issues that need to be improved like more precise inclusion criteria, methods etc.

Thank you for your constructive comments.

Necessary corrections were made in the method section.

Comments on the Quality of English Language

English native speaker should check the manuscript once again due.

English language Edited by MDPI.

Reviewer 3 Report (New Reviewer)

The authors have answered my questions. I think he manuscript has been improved. Thank you.

it is fine. minor changes should be needed. 

Author Response

Dear Editor,

First, we would like thank the reviewers for the helpful comments, which led us to conduct appropriate experiments. The revised manuscript has subsequently been rewritten to address these concerns and comments of the reviewers. English language edited by MDPI.

We are grateful for your understanding and cooperation in this matter.

We believe that the manuscript is now suitable for review. We look forward to your reply.

RESPONSE TO REVIEWERS:

Reviewer 3

Comments and Suggestions for Authors

The authors have answered my questions. I think he manuscript has been improved. Thank you.

Thank you for your constructive comments.

Comments on the Quality of English Language

it is fine. minor changes should be needed.

English language Edited by MDPI.

This manuscript is a resubmission of an earlier submission. The following is a list of the peer review reports and author responses from that submission.

Round 1

Reviewer 1 Report

The manuscript “New generation systemic inflammatory markers for respiratory syncytial virus infection of the lower respiratory tract in children” has conducted a retrospective observational analysis to investigate new immune-inflammatory indices for children with RSV infection. The paper needs extensive English editing and the areas below must be improved:

1.     If the conclusion is that CRP value can be used as a pre-test in RSV, the experiment design needs to include other viral infections as control. How would CRP be used to differentiate RSV v.s. COVID?

2.     What are the advantages of using CRP to determine RSV infection over the gold standard RT-PCR?

3.     The care for newborns with RSV drastically differs from infant, what’s the age distribution for the subjects with RSV+ or RSV- infection? I

4.     Please spell out the acronyms e.g. HGB, HCT, PLT, LYM when they first appear in the manuscript.

5.     Please keep the decimal format consistent. Either use 0.000 or 0,000 in table 1.

6.     Please explain the cut-off in table 3 & 5.

7.     Table 4: what is “Medyan”?

Reviewer 2 Report

The authors studied children who presented to outpatient or emergency clinics with respiratory symptoms and tested for RSV with an immunoassay to determine whether they had RSV. They compared common blood or serum tests to determine whether they, or parameters derived from them--NLR, PLR, SII, aided as a pre-test for diagnosis of RSV.

The authors reported that CRP was the most sensitive indicator of RSV, but that NLR, PLR and SII were also good indicators for RSV primarily in cheldren less than 2 yo and primarily because lymphocyte counts were lower than the RSV- cohort. 

Major criticisms:

1. The authors used immunoassay to diagnose RSV rather than RT-PCR. They state that the immunoassay has 86% sensitivity and 94% specificity without references. Is this from published literature or in-house comparisons. Either way, without a true gold-standard, the specificity and sensitivity of the other tests are inaccurate. It is likely that they would be better, but that is a guess.

2. Since the authors did not test the children for other respiratory viruses, there is no indication that any of the tests are specific for RSV. In fact, there is no reason to believe that would be the case. Without surveying for other viral pathogens, the authors cannot state that the lab tests are indicative of RSV per se. 

3. The essential findings are elevated CRP and depressed lymphocyte count. As the authors state, elevated CRP is also associated with bacterial disease, so it is only informative towards detecting inflammation or infection. In fact, the authors state this in the discussion section with a reference (33). The differences in the NLR, PLR and SII, as the authors acknowledge, are primarily due to higher lymphocyte counts in the younger children. 

4. The discussion section is quite long and covers territory not necessarily relevant to their data. For example Line 264-269 discusses changes in platelet counts. 

Minor comments: check for misspelling: swap on Line 114, specificity on line 119, median in Table 4. 

Reviewer 3 Report

Authors evaluated the relationship between the systemic immune-inflammatory index (SII), neutrophil-to-lymphocyte ratio (NLR) and platelet-to-lymphocyte ratio (PLR) with clinical findings of respiratory syncytial virus (RSV) infection among children with the diagnosis of lower respiratory tract infection (LRTI). They concluded that CRP value was superior to other blood parameters, and the NLR, PLR, and SII index were significantly lower in patients with RSV (+) than RSV (-). However, a review of the study design is necessary to draw conclusions. Appropriate study design and control settings should be used.

 The control setting is inappropriate for this study. As shown in Table 2, the study compares inpatients with RSV infection with outpatients with non-RSV infection. In other words, this study compares severe cases of RSV with mild cases of non-RSV. Comparing inpatients with RSV infection with outpatients with RSV infection may find a new marker for severity assessment.

 Lines 99-100. Lower respiratory tract inflammation is not properly defined. The authors' definition can not distinguish symptoms from upper respiratory tract inflammation. Were there any chest auscultation findings or chest radiograph findings that indicated lower respiratory tract inflammation?

 Line 114: How is first RSV infection determined?

 There is no mention of the limitations of this study.